# Impact of Ventilator Settings on Pulmonary Nodule Localization Accuracy in a Hybrid Operating Room: A Single-Center Study

**DOI:** 10.3390/jcm13175183

**Published:** 2024-09-01

**Authors:** Jiun Yi Hsia, Hsu Chih Huang, Kwong-Kwok Au, Chih Yi Chen, Yu Hsiang Wang

**Affiliations:** 1Division of Thoracic Surgery, Chung Shan Medical University Hospital, Taichung 402, Taiwan; cshy1700@csh.org.tw (J.Y.H.); papuwa924@gmail.com (H.C.H.); speedmoment@hotmail.com (K.-K.A.); cshy1566@csh.org.tw (C.Y.C.); 2School of Medicine, Chung Shan Medical University, Taichung 402, Taiwan; 3Institute of Medicine, Chung Shan Medical University, Taichung 402, Taiwan

**Keywords:** computed tomography, hybrid operating room, pulmonary nodule localization, ventilator settings

## Abstract

**Background:** Pulmonary nodule localization in a hybrid operating room (OR) followed by thoracoscopic operation presents a viable alternative for early lung cancer treatment, potentially supplanting conventional two-stage preoperative computed tomography-guided localization. This hybrid OR technique enables lesion localization under positive ventilation, contrasting with the traditional method requiring concurrent respiratory motion. This study aimed to evaluate our experience with different ventilator settings and the accuracy of pulmonary nodule localization. **Methods:** We retrospectively analyzed 176 patients with multiple pulmonary nodules who had localization procedures in our hybrid operating room. Ninety-five patients were assigned to the traditional ventilator setting group (tidal volume 8–10 mL/kg) and 81 to the lung-protective strategy group (tidal volume < 8 mL/kg). Localization accuracy was assessed via hybrid computed tomography imaging, ensuring that the needle-to-lesion distance was ≤5 mm. Between-group differences were assessed using the chi-squared test, Fisher’s exact test, and the Mann–Whitney U test, as appropriate. **Results:** Pathological findings revealed primary lung malignancy in 150 patients, inclusive of invasive adenocarcinoma, adenocarcinoma in situ, and minimally invasive adenocarcinoma. Multivariate regression analysis identified tidal volume, nodule count, and localization depth as significant predictors of localization accuracy. **Conclusions:** This study demonstrated that ventilator settings with a tidal volume of 8–10 mL/kg significantly enhanced localization accuracy and slightly improved patient oxygenation. However, additional randomized controlled trials are warranted to validate these findings and establish definitive guidelines for future interventions.

## 1. Introduction

The Taiwan Lung Cancer Screening in the Never-Smoker Trial study reported a lung cancer detection rate of approximately 2.6%, with 96.5% of cases diagnosed at stages 0–1, including adenocarcinoma in situ (AIS), minimally invasive adenocarcinoma (MIA), and invasive adenocarcinoma [1]. Thoracoscopic surgery offers a minimally invasive approach for pulmonary nodule resection, significantly improving long-term survival rates [2,3]. However, palpating early lung adenocarcinomas during thoracoscopic surgery is challenging, especially for non-peripheral lesions [4,5]. Consequently, preoperative and intraoperative localization with materials such as micro-coils, hook wires, contrast media, dyes, and fluorescence tracers is essential for successful resection [6]. 

Pulmonary localization procedures are traditionally performed in a computed tomography (CT) suite with patients awake and spontaneously breathing. However, transferring patients to the operating room and inducing general anesthesia introduces risks, including tension pneumothorax, hemothorax, and dislodgment of localization materials [7,8,9]. Thus, performing CT-guided localization in a hybrid operating room, followed by immediate thoracoscopy-assisted resection, has been shown to be both safe and effective, reducing the “risky period” [10,11].

During pulmonary nodule localization in the hybrid operating room (OR), all patients receive general anesthesia with double-lumen endotracheal intubation, and the localization is carried out at the same end-inspiratory phase. Ventilator settings directly impact localization accuracy and oxygen saturation levels. For patients requiring one-lung ventilation during thoracoscopic surgery under general anesthesia, anesthesiologists typically use a lung-protective strategy with low tidal volume in volume control mode. Literature suggests using low tidal volume ventilation (6–8 mL/kg) with 5 cm H_2_O positive end-expiratory pressure to reduce postoperative pulmonary complications [12,13]. 

However, this setting might not be optimal for pulmonary nodule localization in the hybrid OR, as low tidal volumes can lead to rapid atelectasis progression during apnea at the end-inspiratory phase, resulting in hypoxia and reduced localization accuracy. Additionally, respiratory rates may reach 12–18 breaths/min to maintain adequate minute ventilation in low tidal volume settings, resulting in an inspiration phase of approximately 1 s. These short inspiration phases can cause synchronization issues between the anesthesiologist and surgeon, leading to inaccuracies.

Traditionally, high tidal volumes (10 mL/kg) reduce atelectasis and improve saturation, though this has not been confirmed under modern general anesthesia. Using these settings, it is recommended that respiratory rates be eight to ten breaths per minute, allowing better synchronization of the inspiration hold between the anesthesiologist and surgeon. We hypothesized that traditional ventilator settings enhance oxygenation, minimize the need for lung reinflation during localization of multiple pulmonary ground-glass nodules, and improve localization accuracy. This study aimed to assess our experience with various ventilator settings and their impact on pulmonary nodule localization accuracy.

## 2. Materials and Methods

We conducted a retrospective review of patient data from individuals who underwent preoperative localization of multiple pulmonary partially solid ground-glass nodules in the hybrid operating room at Chung Shan Medical University Hospital between 1 January 2022, and 31 December 2022. The study received formal approval from the Ethics Committee of Chung Shan Medical University Hospital (approval number CS2-22172). All procedures were carried out in full compliance with relevant laws, institutional guidelines, and ethical standards to maintain the integrity and safety of the research. The inclusion criteria were as follows: (1) age ≥ 18 years, (2) low-risk classification according to the American College of Chest Physicians perioperative surgical risk evaluation based on lung function (forced expiratory volume and diffusion capacity of carbon monoxide ≥ 80% predicted), and (3) a diagnosis of multiple partial solid pulmonary ground-glass nodules requiring localization in the hybrid OR.

A total of 176 patients with multiple pulmonary partial solid ground-glass nodules who underwent localization in the hybrid OR were included. Among them, 95 were assigned to the traditional ventilator setting group (tidal volume 8–10 mL/kg predicted body weight) and 81 to the lung-protective strategy ventilator setting group (tidal volume < 8 mL/kg predicted body weight). Ventilator settings were chosen according to the preferences of the surgeon and anesthesiologist, with the details recorded in the anesthesia documentation. Details regarding patient inclusion criteria, exclusion criteria, and classification are illustrated in Figure 1.

Prior to the localization procedure, all patients received double-lumen endotracheal tube intubation and anesthesia induction. Patients were positioned in either the decubitus or 30° lateral tilt position. Ventilation was conducted using pressure control and volume guarantee mode on anesthesia workstations (Aisys CS2™, General Electric Healthcare, Chicago, IL, USA). Tidal volume settings were divided into two groups: the traditional group (8–10 mL/kg predicted body weight) and the lung-protective strategy group (<8 mL/kg predicted body weight). Respiratory rate settings were adjusted based on minute ventilation and body surface area (BSA) to prevent hypoventilation. During localization, the fraction of inspired oxygen for all patients was set at 50%. The hybrid localization system used a C-arm cone beam CT (ARTIS Pheno^®^; Siemens Healthcare GmbH, Erlangen, Germany) to gather precise data for pulmonary nodule localization, inclusive of the distance from the chest wall to the nodule and its characteristics. Oxygen saturation levels were continuously monitored through pulse oximetry using a sensor placed on the contralateral fingernail bed, with the readings displayed on both the anesthesia workstations and the localization monitor. Three experienced surgeons collaborated on all localization procedures. One surgeon conducted the localization; another managed respiration holds; and the third surgeon, working with a radiologist, handled lesion recognition and assessed localization accuracy. Following localization, patients underwent thoracoscopic surgery in the hybrid OR, with ventilator settings promptly adjusted to a lung-protective strategy by the anesthesiologist. If the resected nodule was identified as AIS or at least AIS in the frozen-section analysis, no further lung resection was performed, and the incision was closed. However, if invasive adenocarcinoma or another histological malignancy was detected, lesion size was assessed, and the option of proceeding with lobectomy or segmentectomy was considered. No pre-localization recruitment procedure was implemented, although previous studies suggest that it could enhance accuracy and oxygenation [14]. The complete algorithm for localizing multiple pulmonary partial solid ground-glass nodules and subsequent surgery is depicted in Figure 2.

Re-inflation was performed if pulse oximetry indicated a saturation level below 90% (oxygen saturation [SpO_2_] < 90). The apnea time was documented by the anesthesiologist and calculated from the start of localization procedure until successful completion of the procedure without SpO_2_ dropping below 90%. Localization accuracy was assessed using hybrid CT imaging, confirming that the distance between the localization needles and the lesion remained within 5 mm in a 3D view. Two radiologists and one experienced surgeon conducted an immediate post-procedure review of the images. Secondary outcomes measured included apnea duration, oxygen saturation, re-inflation rates, procedure length, and the occurrence of pneumothorax. 

The chi-squared test, Fisher’s exact test, and the Mann–Whitney U test were employed as appropriate to assess differences between groups. Multivariate logistic regression was used to analyze localization accuracy and re-inflation rates, with the outcomes reported as odds ratios. A *p*-value of less than 0.05 was considered statistically significant. All statistical analyses were performed using IBM SPSS Statistics (version 25) and Microsoft Excel (Microsoft Corporation, Seattle, WA, USA).

## 3. Results

A total of 176 patients underwent localization of multiple pulmonary ground-glass nodules in our hospital’s hybrid operating room. The details of patient characteristics are presented in Table 1. Among them, 95 patients had localization with a tidal volume of 8–10 mL/kg and 81 experienced localizations with a tidal volume of <8 mL/kg. Most patients had two pulmonary lesions (N = 120, 68.2%), while 36 had three lesions (N = 36, 20.5%), and 20 had four lesions (N = 20, 11.4%). To ensure a safe resection margin, the choice of localization instrument was determined by the thoracic surgeon, and 57% (N = 101) of patients underwent localization using hook wires. No notable differences were found between the groups regarding age; gender; height; weight; or preoperative pulmonary function metrics, including functional vital capacity, forced expiratory volume in one second, and lung diffusing capacity for carbon monoxide. 

Most patients underwent wedge resection (N = 159, 90.3%) after frozen-section analysis confirmed a pure lepidic pattern of the lesions. However, three patients in each group (tidal volume < 8 mL/kg and 8–10 mL/kg) underwent segmentectomy after a frozen-section pathologic diagnosis of lung malignancy to ensure an adequate resection margin. Additionally, five patients in the < 8 mL/kg group and six in the 8–10 mL/kg group underwent lobectomy to ensure an adequate resection margin. No hook wire dislodgment occurred during lobectomy, and there was no significant difference in the type of surgery between the groups.

Final pathology reports revealed primary pulmonary malignancy in 150 cases (85.2%), including invasive adenocarcinoma, AIS, and MIA. Adenocarcinomas were predominantly clinical-stage T1a (<1 cm) or T1b (>1 cm but <2 cm), with no progression to clinical stage T1c. Notably, one case involved secondary pulmonary malignancy, a metastasis from a colorectal adenocarcinoma. Additionally, 25 patients had benign pathology reports indicating only fibrosis or chronic inflammation-related changes in the lung. Nevertheless, the pathology reports showed no meaningful differences between the two groups.

The procedure time was 18 min for the <8 mL/kg group and 16 min for the 8–10 mL/kg group, with a significant difference (Table 2, *p* = 0.032). This difference may be attributed to the higher reinflation rate and longer apnea time in the <8 mL/kg group than in the 8–10 mL/kg group, although this difference was not significant (43.2% vs. 32.6%). The 8–10 mL/kg group had significantly better oxygen saturation after localization apnea than the <8 mL/kg group (90% vs. 88%, *p* = 0.006). Additionally, the localization accuracy was significantly higher in the tidal volume 8–10 mL/kg group than in the tidal volume <8 mL/kg group (90.5% vs. 51.9%, *p* = 0.001). Forty-three patients had iatrogenic pneumothorax after localization (N = 43, 24.4%), none of whom required immediate thoracentesis for procedure completion (Table 2).

Unilateral logistic regression logistic regression analysis indicated that body mass index (BMI), tidal volume group, number of lesions, and localization depth were linked to localization accuracy, with odds ratios of 0.81, 0.11, 0.50, and 0.95, respectively. Multivariate logistic regression analysis determined that tidal volume, number of lesions, and localization depth were significant predictors of localization accuracy, with Wald’s test showing *p* < 0.001. The odds ratios for these variables were 0.79, 0.08, 0.48, and 0.96, respectively (Table 3).

## 4. Discussion

To the best of our knowledge, this study is the first to investigate the effects of various ventilator settings on pulmonary nodule localization in hybrid operating rooms under general anesthesia. Unlike the traditional two-stage localization in CT rooms (preoperative CT, POCT), one-stage localization in hybrid ORs (intraoperative CT, IOCT) involves positive pressure ventilation rather than spontaneous breathing. Consequently, adjusting respiratory rate, tidal volume, and pressure is crucial for accurate pulmonary nodule localization. Current evidence and clinical guidelines recommend a lung-protective strategy, including low tidal volume (<6–8 mL/kg predicted body weight), plateau pressure (<28–30 cm H_2_O), and driving pressure (<14 cm H_2_O) for managing acute respiratory distress syndrome and during general anesthesia [15,16,17]. Our study, however, revealed that this lung-protective approach utilizing low tidal volume could potentially diminish localization accuracy. 

During the localization process, regional atelectasis rapidly develops once apnea begins due to oxygen consumption and diaphragm movement, leading to uneven lung collapse and potentially compromising procedural accuracy. Research indicates that atelectasis is influenced by the patient’s BMI and age [15,17,18]. In patients with obesity, these effects are more pronounced and rapid due to diaphragmatic limitation and muscle weakness, resulting in shorter apnea times and faster atelectasis progression [19]. This limitation was also evident in this study, where approximately 56 overweight patients (BMI > 25.0, N = 28 in each group) were included. Both univariate and multivariate logistic regression analyses confirmed that BMI negatively impacted localization accuracy. 

For ventilator settings during localization, minute ventilation is estimated to avoid hypoventilation [20], primarily based on metabolic rate. Direct metabolic rate assessment via calorimetry is feasible; however, it is not widely accessible, making BSA a practical substitute for estimating metabolic rate and minute ventilation. Minute ventilation in patients with normothermia is three to four times that of BSA. Consequently, with a lung-protective strategy of <8 mL/kg, the respiratory rate typically ranges from 12 to 18 breaths/min to achieve adequate minute ventilation. A standard inspiration/expiration ratio for most situations is 1:2, with the inspiration phase lasting about 1.5 s. However, consistent tidal volume inspiration between planning and localization is crucial for accuracy. Thus, 1.5 s may be too brief for the anesthesiologist and thoracic surgeon to synchronize inspiration holds, potentially disrupting the localization plan and reducing accuracy. When ventilator settings are increased to >10 mL/kg, the respiratory rate is typically adjusted to 8–10 breaths/min, extending the inspiration phase to 2–2.5 s and facilitating synchronization between the anesthesiologist and thoracic surgeon during inspiration holds (Figure 3). The <8 mL/kg group exhibited significantly lower accuracy than the 8–10 mL/kg group (Table 3), indicating that the “protective effect” of the lung-protective strategy may impede accurate localization, necessitating re-puncture. 

An alternative approach to address synchronization challenges during inspiration holds is to perform the whole localization procedure without respiration holds. However, some electromagnetic navigation-guided localization software records target movement during continuous breathing cycles, with variations ranging from 5 mm to 18 mm between breaths [21]. This motion discrepancy can exceed the lesion size, leading to localization errors. Moreover, concerns also exist about whether this ventilator setting could damage lung parenchyma. Nonetheless, after localization, ventilator settings are promptly adjusted to a lung-protective strategy in this study, and no postoperative pulmonary complications were reported during subsequent care.

The number of lesions also significantly affects localization accuracy. During localization, the inspiration hold causes rapid alveolar oxygen consumption at 250 mL/min, while carbon dioxide is expelled at 20 mL/min, contributing to lung atelectasis [22]. Based on our experience with the localization of single pulmonary ground-glass nodules, the apnea duration is typically kept under 2 min, with the majority of patients not experiencing atelectasis or hypoxia. However, for multiple ground-glass nodule localization, apnea may extend to approximately 5–6 min, increasing oxygen consumption and atelectasis risk. As a result, localization accuracy declines with more lesions, particularly in the tidal volume 8–10 mL/kg group (Table 3). 

Research indicates no difference in lung atelectasis between low and conventional tidal volumes during positive ventilation [23]. However, pulmonary nodule localization in the hybrid OR requires a breath-hold without ventilation, challenging the notion that the lung-opening strategies are entirely invalid. The tidal volume 8–10 mL/kg group had lower end oxygenation (SpO_2_: 88% vs. 90%, respectively; *p* = 0.006) and a slightly higher reinflation rate (43.2% vs. 32.6%) than the <8 mL/kg group. Multivariate logistic regression also showed that lesion number negatively affected localization accuracy (Table 3, odds ratio = 0.48, *p* = 0.014). To mitigate this, a pre-procedure recruitment maneuver for full lung expansion demonstrated improved oxygenation and accuracy in multiple pulmonary nodule localization [14], and another study underscored the safety and effectiveness of ventilating one lung during localization.

Although this study addresses ventilator settings and pulmonary nodule localization in a hybrid OR, it has some limitations. Firstly, it was not a randomized controlled trial and ventilator settings were based on anesthesiologists’ and surgeons’ preferences without any specific selection criteria. Nevertheless, no randomized controlled trials or large-scale studies have addressed ventilator settings and pulmonary nodule localization in hybrid ORs.

Secondly, our results reflect nearly a decade of refining a pulmonary nodule localization team. Since 2018, our thoracic surgery team has used traditional CT for partially solid pulmonary nodule localization, transitioning to IOCT in early 2022. To date, our team has performed over 3500 pulmonary nodule localization operations. The direct clinical implications of using hybrid ORs without traditional CT for localization may take time to fully understand. Recently, Chang Gung Memorial Hospital conducted a small randomized control trial comparing IOCT and POCT, which showed reduced procedural time and radiation exposure but did not address localization accuracy [24]. The study aimed to identify the best methods for pulmonary nodule localization to achieve limited resection with adequate margins in wedge resection and segmentectomy. We are conducting a large-scale retrospective study to directly compare IOCT and POCT, providing further insights into their comparative effectiveness. Thirdly, none of the patients in our study had obstructive or restrictive lung disease. Whether our results apply to these more vulnerable patients should be discussed with anesthesiologists considering individual clinical circumstances. Moreover, further studies should include this specific group. Fourthly, this study was retrospective, with an imbalanced division of patients into different tidal volume groups. While no unmeasured confounding variables could have influenced the results in terms of patient characteristics, allocation bias may have influenced the results. Consequently, a prospective study with a carefully structured protocol is needed to confirm and validate our results. Lastly, our study involved relatively healthy patients without cardiovascular or other systemic diseases. Thus, this study offers only initial evidence of the benefits of the 8–10 mL/kg tidal volume setting for relatively healthy patients undergoing localization of multiple partially solid ground-glass nodules in a hybrid operating room. Future research should investigate the effects of ventilator settings in a larger patient cohort through randomized controlled trials, focusing on assessing the safety of the procedure for individuals at elevated risk.

## 5. Conclusions

This retrospective observational research suggests that the 8–10 mL/kg tidal volume ventilator setting effectively increases localization accuracy and slightly improves patient oxygenation. Factors such as patient BMI, tidal volume, number of lesions, and localization depth were significant predictors of localization accuracy. However, additional randomized controlled studies are necessary to confirm our findings and develop decision criteria for future procedures.

## Figures and Tables

**Figure 1 jcm-13-05183-f001:**
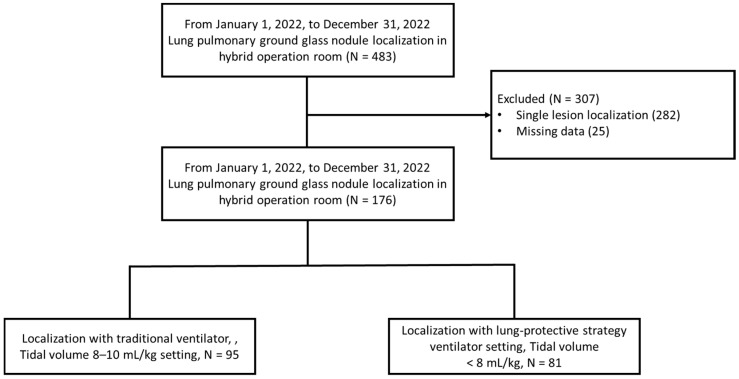
Flowchart of patient selection and classification.

**Figure 2 jcm-13-05183-f002:**
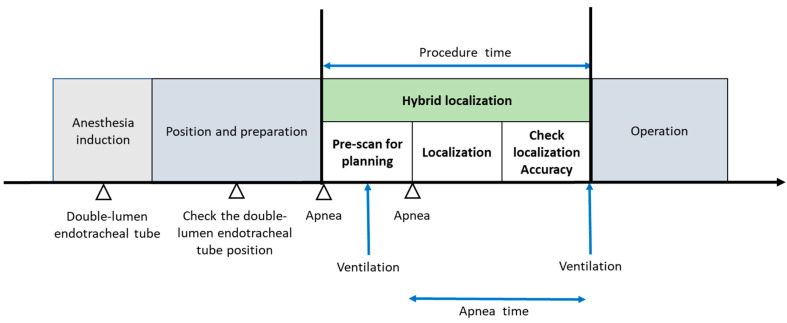
Phases of multiple pulmonary ground-glass nodule localization and tumor excision operation.

**Figure 3 jcm-13-05183-f003:**
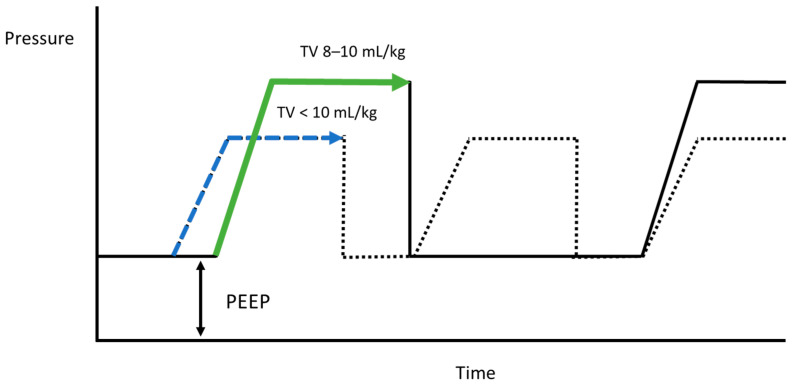
Comparison of peak pressure and inspiration time between the low- and high-tidal-volume groups, indicated by the blue and green dotted pressure–time graphs, respectively.

**Table 1 jcm-13-05183-t001:** Characteristics of study participants.

		<8 mL/kg N = 81	8–10 mL/kg N = 95	*p* Value
Gender	Female	62 (76.5%)	78 (82.1%)	0.362 †
	Male	19 (23.5%)	17 (17.9%)	
Age, median (IQR)		57 (51–65)	54 (47–65)	0.375
Height median (IQR)		159.0 (154.0–165.0)	159.0 (155.0–165.0)	0.898
Body weight, median (IQR)		59.0 (52.0–64.0)	58.5 (54.0–68.0)	0.707
BMI median (IQR)		22.4 (21.3–24.3)	23.1 (21.4–25.3)	0.098
PBW, median (IQR)		54.2 (48.8–63.3)	51.5 (48.8–58.8)	0.132
Tidal volume median (IQR)		450 (458–500)	550 (500–625)	<0.001
Contralateral lung operation	No	65 (80.2%)	77 (81.1%)	0.893 †
	Yes	16 (19.8%)	18 (18.9%)	
Localization position	30 degree tilt	64 (79.0%)	72 (75.7%)	0.611
	Decubitus	17 (21.0%)	23 (24.2%)	
Lung function	FVC	102.0 (93.0–107.0)	101.0 (93.0–112.0)	0.260
	FEV_1_	99 (88.0–107.0)	98.0 (90.0–107.0)	0.501
Lung function	DLCO	95.0 (87.0–108.0)	98.0 (87.0–105.0)	0.547
Lesions	2	55 (67.9%)	65 (68.4%)	0.115 †
	3	13 (16.0%)	23 (24.2%)	0.547
Lesions	4	13 (16.0%)	7 (7.4%)	0.115 †
Size (mm, median IQR)	7.30 (6.0–8.50)	7.20 (5.60–8.80)	0.673	
Depth (mm, median IQR)	70.0 (55.0–80.0)	65.0 (55.0–75.0)	0.209	
Dye or hook	Dye	31 (38.3%)	44 (46.3%)	0.281 †
Depth (mm, median IQR)	Hook	50 (61.7%)	51 (53.7%)	
Operation	Wedge	73 (90.1%)	86 (90.5%)	0.98 †
	Segmentectomy	3 (3.7%)	3 (3.2%)	
Operation	Lobectomy	5 (6.2%)	6 (6.3%)	0.98 †
Pathology	AIS	11 (13.6%)	18 (18.9%)	0.70 †
	MIA	43 (53.1%)	46 (48.4%)	
Pathology	Adenocarcinoma	15 (18.5%)	17 (17.9%)	0.70 †
	Metastasis	1 (1.2%)	0	
	Benign	11 (13.6%)	14 (14.7%)	

BMI, body mass index; AIS, adenocarcinoma in situ; MIA, minimally invasive adenocarcinoma; IQR, interquartile range; FVC, forced vital capacity; FEV_1_, forced expiratory volume in one second; DLCO, diffusion capacity of carbon monoxide; PBW, predicted body weight; † Fisher’s exact test/chi-square test; Mann–Whitney U test.

**Table 2 jcm-13-05183-t002:** Comparison between patient groups.

		<8 mL/kg N = 81	8–10 mL/kg N = 95	*p* Value
Apnea time		5.25 (4.30–6.50)	5.05 (4.25–5.80)	0.091
SpO_2_ (%, median, IQR)		88 (84–94)	90 (87–97)	0.006
Procedure time (min, IQR)		18.0 (15.0–20.0)	16.0 (14.0–19.0)	0.032
Accuracy (within 5 mm)		42 (51.9%)	86 (90.5%)	0.001
Accuracy with Lesions	2	33 (60.0%)	60 (92.3%)	
	3	6 (46.2%)	21 (91.3%)	
	4	3 (23.1%)	5 (71.4%)	
Re-inflation	No	46 (56.8%)	64 (67.4%)	0.149 †
	Yes	35 (43.2%)	31 (32.6%)	
Pneumothorax	No	57 (70.4%)	76 (80%)	0.138 †
	Yes	24 (29.6%)	19 (20%)	

Min, minutes; IQR, interquartile range; † Fisher’s exact test/chi-square test; Mann–Whitney U test.

**Table 3 jcm-13-05183-t003:** Logistic regression analysis of factors influencing localization accuracy.

Accuracy	Univariate Logistic Regression Analysis		Multivariate Logistic Regression Analysis	
	Odds Ratio (95% CI)	*p* Value	Odds Ratio (95% CI)	*p* Value
BMI	0.81 (0.72–0.91)	0.000	0.79 (0.68–0.92)	0.003
Tidal volume/PBW 8–10 mL/kg or <8 mL/kg	0.11 (0.05–0.25)	0.000	0.08 (0.03–0.21)	0.000
Lesion number	0.50 (0.32–0.80)	0.003	0.48 (0.27–0.86)	0.014
Dye or hook	1.19 (0.60–2.33)	0.619		
Size (mm, median IQR)	1.01 (0.88–1.16)	0.875		
Depth (mm, median IQR)	0.95 (0.93–0.98)	0.000	0.96 (0.94–0.99)	0.018

CI, confidence interval; BMI, body mass index; IQR, interquartile range; PBW, predicted body weight.

## Data Availability

The datasets used and/or analyzed during the current study are available from the corresponding author on reasonable request.

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
