# Peer review of "Impact of Ventilator Settings on Pulmonary Nodule Localization Accuracy in a Hybrid Operating Room: A Single-Center Study"

_jcm, 2024, doi:10.3390/jcm13175183_

Round 1

Reviewer 1 Report

Comments and Suggestions for Authors

Tumor localization is important for accurate resection of small GGOs. In this study, the authors involved a total of 176 patients with multiple pulmonary nodules who underwent localization 92 in the hybrid OR were included. Among them, 95 were assigned to the traditional ventilator setting group (tidal volume 8–10 mL/kg predicted body weight) and 81 to the protective lung strategy ventilator setting group (tidal volume < 8 mL/kg predicted body weight). The data suggests that the tidal volume 8–10 mL/kg ventilator setting effectively increases localization accuracy and slightly improves patient oxygenation. These findings are interesting and clinical significant. However, some issues need to be addressed to improve the paper.

1. For group < 8 mL/kg N = 81, the localization accuracy is too low as compared with previous other studies. For example, the localization accuracy for 2 lesions is 60% vs 90%, for 4 lesions is 23% vs 71%.  Why such a big difference between groups? Why such a low accuracy in low tidal volume group? 

2. What about the operators for localization? How many operators are involved in the procedure? Is there any difference for he operators between groups? This is important for data analysis and discussion.

Author Response

Dear reviewer 
Thank you for your comments on our study. Here are my responses to your observations:
First, our study follows the most stringent accuracy standards available in the literature, permitting only a 5 mm error in three-dimensional analysis. The literature presents various definitions of "successful targeting rate," with localization accuracy ranging from 53% to 98% [1]. Some studies consider technical success as localization within the same segment, which may be deemed inaccurate according to our criteria [2]. Additionally, our anesthesiologists use a 90% saturation threshold to determine the end of apnea. In the low tidal volume group, as indicated in Table 1, the tidal volume difference between groups is 100 ml. Based on lung reabsorption physiology during apnea, the apnea time in the low tidal volume group may be 30 seconds shorter than in the 8-10 mL/kg group. While there was no statistically significant difference in re-inflation rates between groups (N = 35, 43.2% vs. N = 31, 32.6%), a trend toward higher re-inflation rates with smaller tidal volumes was observed. Re-localization after re-inflation may vary, impacting localization accuracy. With 3-4 lesions, the increased localization time and risk of re-inflation can exacerbate these issues, leading to greater localization discrepancies as the number of lesions increases.
Second, in our study, despite the involvement of three qualified surgeons, we implemented specific roles to minimize operational bias. One surgeon performed the localization procedure, while another was responsible for controlling respiration holds. The third surgeon, in collaboration with a radiologist, handled lesion recognition and assessed localization accuracy. This approach was designed to minimize bias related to the operation. We have added this explanation to the “Materials and Methods” section (116-118).
Sincerely 
Dr. Yu Hsiang Wang

Reference: 
1. Cornella KN, Repper DC, Palafox BA, Razavi MK, Loh CT, Markle KM, Openshaw LE. A Surgeon's Guide for Various Lung Nodule Localization Techniques and the Newest Technologies. Innovations (Phila). 2021 Jan-Feb;16(1):26-33. doi: 10.1177/1556984520966999. Epub 2020 Oct 30. PMID: 33124923.
2. Seo JM, Lee HY, Kim HK, Choi YS, Kim J, Shim YM, Lee KS. Factors determining successful computed tomography-guided localization of lung nodules. J Thorac Cardiovasc Surg. 2012 Apr;143(4):809-14. doi: 10.1016/j.jtcvs.2011.10.038. Epub 2011 Nov 20. PMID: 22104686.

Reviewer 2 Report

Comments and Suggestions for Authors

Authors wrote a nice paper about a very actual issue in the field of thoracic surgery, because use of hybrid operating room is rapidly expanding.

Surely the retrospective design is a major drawback, but the next prospective studies will increase he knowledge in this field.

I have one question: how did the surgeons choose the kind of ventilator setting? what were the criteria and how these criteria will be selected in a future study?

Author Response

Dear reviewer 
Thank you for your comments on our study. Here are my responses to your observations:
To date, there is no high-quality evidence regarding optimal ventilator settings during the localization procedure. The only principle our localization team has followed, after performing over 1,000 cases since 2022, is safety. Initially, we used volume control ventilation (VCV mode) for the localization procedure. However, VCV mode exhibited a rapid pressure-volume curve without a decelerating flow pattern compared to pressure control mode. We transitioned to Pressure Control with Volume Guarantee mode (PCV-VG) after consulting with our anesthesiologists. Cooperation between anesthesiologists and surgeons is crucial.
In my opinion, current localization methods are limited by the tools available. The most convenient approach for localizing pulmonary partial solid ground-glass nodules is to perform the procedure with one-lung ventilation. Single peripheral lesions can be easily localized with a needle-dye system even with one-lung ventilation. However, challenges arise with central or multiple lesions. Localization of central lesions requires hook-wire or microcoils systems, which are difficult to use with one-lung ventilation due to respiratory variations. Additionally, in cases with multiple lesions, iatrogenic pneumothorax may develop during the localization, potentially leading to failure. We are still exploring ventilator settings that could facilitate localization procedures in the hybrid operating room.
Sincerely 
Dr. Yu Hsiang Wang

Reviewer 3 Report

Comments and Suggestions for Authors

Attached is my comments to the authors.

Author Response

Dear reviewer
Thanks for your comment for out study. Here is my reply to your comments. 
First, the artis pheno system had its limitations in range of scan, which measured in 9*17 cm in size. Once the patient had two or more lesion with anterior chest and posterior chest, the 30° tilt position may be unable to access the posterior lesion. Therefore, most of the patients have their localization procedure in decubitus position. In our experience, the lesions located at the dorsal side are easier to access, because of the intercostal distance are more widening and there were no breast and axillary soft tissue bothering the localization procedure. We had listed the localization position in our Table 1. Characteristics of study participants. There was no statistical difference between groups (P = 0.611) Second, all our patients have localization with pressure control and volume guarantee mode on anesthesia workstations (Aisys CS2™, General Electric Healthcare, Chicago, IL). It’s a pity that we had no peak inspiratory pressure record in our anesthesia workstations database. However, in our localization experience, we tend to have lower peak inspiration pressure around 15 cmH2O. Once a higher peak inspiration pressure, we would advise our anesthesiologist to check the double lumen endotracheal tube first.
Sincerely 
Dr. Yu Hsiang Wang